# Ranolazine Counteracts Strength Impairment and Oxidative Stress in Aged Sarcopenic Mice

**DOI:** 10.3390/metabo12070663

**Published:** 2022-07-18

**Authors:** Alessio Torcinaro, Donato Cappetta, Francesca De Santa, Marialucia Telesca, Massimiliano Leigheb, Liberato Berrino, Konrad Urbanek, Antonella De Angelis, Elisabetta Ferraro

**Affiliations:** 1Institute of Biochemistry and Cell Biology (IBBC), National Research Council of Italy (CNR), Via Ercole Ramarini, 32, Monterotondo, 00015 Rome, Italy; alessio.torcinaro@ibbc.cnr.it (A.T.); francesca.desanta@cnr.it (F.D.S.); 2Istituto Dermopatico dell’Immacolata (IDI), Istituto di Ricovero e Cura a Carattere Scientifico (IRCCS), Experimental Immunology Laboratory, Via Monti di Creta, 104, 00167 Rome, Italy; 3Department of Experimental Medicine, Division of Pharmacology, University of Campania “Luigi Vanvitelli”, 80138 Naples, Italy; donato.cappetta@unicampania.it (D.C.); marialucia.telesca@unicampania.it (M.T.); liberato.berrino@unicampania.it (L.B.); antonella.deangelis@unicampania.it (A.D.A.); 4Orthopaedics and Traumatology Unit, “Maggiore della Carità” Hospital, Department of Health Sciences, University of Piemonte Orientale (UPO), 28100 Novara, Italy; massimiliano.leigheb@uniupo.it; 5Department of Molecular Medicine and Medical Biotechnologies, University of Naples Federico II, 80138 Naples, Italy; urbanek@ceinge.unina.it; 6CEINGE-Advanced Biotechnologies, 80138 Naples, Italy; 7Department of Biology, University of Pisa, 56126 Pisa, Italy

**Keywords:** aging, skeletal muscle, sarcopenia, metabolic reprogramming, ranolazine

## Abstract

Sarcopenia is defined as the loss of muscle mass associated with reduced strength leading to poor quality of life in elderly people. The decline of skeletal muscle performance is characterized by bioenergetic impairment and severe oxidative stress, and does not always strictly correlate with muscle mass loss. We chose to investigate the ability of the metabolic modulator Ranolazine to counteract skeletal muscle dysfunctions that occur with aging. For this purpose, we treated aged C57BL/6 mice with Ranolazine/vehicle for 14 days and collected the *tibialis anterior* and *gastrocnemius* muscles for histological and gene expression analyses, respectively. We found that Ranolazine treatment significantly increased the muscle strength of aged mice. At the histological level, we found an increase in centrally nucleated fibers associated with an up-regulation of genes encoding MyoD, Periostin and Osteopontin, thus suggesting a remodeling of the muscle even in the absence of physical exercise. Notably, these beneficial effects of Ranolazine were also accompanied by an up-regulation of antioxidant and mitochondrial genes as well as of NADH-dehydrogenase activity, together with a more efficient protection from oxidative damage in the skeletal muscle. These data indicate that the protection of muscle from oxidative stress by Ranolazine might represent a valuable approach to increase skeletal muscle strength in elderly populations.

## 1. Introduction

Aging is characterized by an accumulation of cellular alterations leading to a progressive functional decline. Common cellular hallmarks of aging include genomic instability, cell senescence, reduced responsiveness to environmental cues, stem cell exhaustion, oxidative stress and mitochondrial dysfunctions—the connections among which are not entirely clear [1].

In skeletal muscles, physiological aging leads to a typical decrease in muscle mass and strength, defined as sarcopenia. Sarcopenia is a progressive and multifactorial disorder associated with many adverse outcomes such as bone fragility, fractures, disability and the consequent socio-economic issues [2]. Muscle strength reduction is considered a more reliable predictive outcome for sarcopenia than muscle mass decrease [3]. Loss of muscle mass is mainly a consequence of a reduced ratio between protein synthesis and protein breakdown, which leads to a lower myofiber cross-sectional area (CSA). In addition, myofiber death and reduced myogenic capability, along with muscle stem cell reservoir exhaustion, contribute to skeletal muscle atrophy associated with a myofiber number lowering during aging [4,5,6].

A major factor contributing to age-dependent muscle degeneration is the disturbance of mitochondrial function including a decline in mitochondrial mass, Krebs cycle enzymes, oxidative phosphorylation and ATP synthesis. Mitochondrial dysfunction reduce muscle strength and could likely have an impact on protein synthesis and loss of muscle mass [7,8,9,10,11,12]. In addition, a reduction in physical activity occurring during aging impairs mitochondrial biogenesis and protein synthesis. Aged muscles also display alterations in myofiber type ratio, with the fast-twitch (type II) glycolytic myofibers being more vulnerable to atrophy—thus decreasing in number—than the oxidative (type I) ones [13,14,15,16,17].

Another typical hallmark of sarcopenic muscles is the severe oxidative stress which is unable to be counteracted due to insufficient antioxidant enzyme activity [18]. This is tightly connected with the bioenergetic failure through a vicious cycle, since mitochondria are both the major source of reactive oxygen species (ROS) and a primary target—particularly mitochondrial DNA (mtDNA)—of oxidative stress. Indeed, mitochondrial turnover and quality control—including mitophagy, mitochondrial dynamics (fusion/fission) and biogenesis—are altered in aged muscles, thus contributing to the age-related accumulation of damaged proteins and increased ROS production which, in turn, affect mtDNA, mitochondrial function and muscle homeostasis [19,20]. For all these reasons, the regulation of muscle metabolism could potentially ameliorate the compromised phenotype in sarcopenia.

The piperazine derivative Ranolazine is a metabolic modulator able to inhibit free fatty acid β-oxidation, thus fostering glucose consumption and improving glucose oxidation. Indeed, switching the source of acetyl-CoA from fatty acids toward pyruvate results in a greater ATP yield. Ranolazine is a well-tolerated drug and has been approved for the management of cardiac dysfunctions, particularly for chronic stable angina pectoris. Its protective effect also encompasses the ability to be a selective inhibitor of late Na^+^ inward current [21,22,23,24,25,26]. Another metabolic modulator of β-oxidation, Trimetazidine, was shown to nurture muscle integrity and improve muscle strength in aging in both humans and mice [27,28,29,30]; hence, we hypothesized that the Ranolazine-mediated optimization of energy might also be beneficial in sarcopenic skeletal muscle by counteracting mitochondrial dysfunction.

In order to study the effect of Ranolazine in aging and its ability to counteract sarcopenia, we treated the 24-month-old mice with this metabolic modulator and we found that it enhances muscle strength and increases the number of centrally nucleated myofibers. Moreover, we observed an up-regulation of the myogenic gene *Myod1,* of key mitochondrial genes as well as a robust reduction in ROS and RNS in skeletal muscle.

## 2. Results

### 2.1. Muscle Performance of Aged Mice Improves upon Ranolazine Treatment

To assess the effect of Ranolazine on muscle functionality during aging, we performed the inverted screen test [28,31] in 24-month-old C57BL/6 male mice treated with Ranolazine for 14 consecutive days (Figure 1A). Based on a prior analysis of old mice strength [28], a sample size, estimated by the G*Power software (version 3.1.9.2.; Düsseldorf, Germany), of a minimum of five animals per group was calculated to be required to obtain significant results. By measuring the longest suspension time on the upside-down screen against gravity, this test allows the evaluation of the muscle strength of each mouse. We found that Ranolazine treatment significantly increases the muscle strength compared to old untreated mice (Figure 1B). Indeed, the latency to fall down, expressed as the longest suspension time, increases from 30.36 ± 28.38 to 81.53 ± 28.82 (*p* < 0.05) after 14 days of Ranolazine treatment. We recently showed that young (3-month-old) C57BL/6 male mice exhibit a 2-fold increase in strength compared to old mice (24-month-old mice) [32]. Therefore, we can conclude that Ranolazine administration leads to a significant improvement in muscle functionality in aged mice, approaching the values typical of young mice.

### 2.2. Effect of Ranolazine on Body Weight, Muscle Mass and Centrally Nucleated Myofibers of Old Mice

In order to evaluate the effect of Ranolazine on skeletal muscle mass, we measured the body weight of mice and we observed that Ranolazine-treated mice showed a trend towards a decreasing total body weight compared to untreated mice (Figure 1C). Vice versa, the measurement of a specific muscle weight revealed an increase in TA weight and a trend towards an increasing GSN weight in Ranolazine-treated mice compared to untreated mice (Figure 1D,E).

To analyze the effect of Ranolazine on skeletal muscle in greater detail, the cross-sectional area (CSA) of myofibers and myofiber size distribution were also evaluated on TA muscle cryosections of both Ranolazine-treated and untreated mice. In order to perform this analysis, we stained TA cryosections with an antibody against Laminin α1, a basal lamina component that localizes around each myofiber, and we used the DAPI (4′,6-diamidino-2-phenylindole) dye to counterstain nuclei (Figure 2A). While the mean of myofiber CSA did not reveal any change between Ranolazine-treated mice and untreated ones (Figure 2B), a deeper analysis by the frequency histograms revealed a slight increase in the number of larger myofibers in Ranolazine-treated mice compared to control mice (Figure 2C) which could suggest an attempt of size recovery upon Ranolazine treatment. This is a different trend compared to that induced by the metabolic modulator Trimetazidine [28].

We also evaluated the expression of genes found to be upregulated in muscle growth [33]. Among them, Ranolazine induces an increase in *Postn* encoding Periostin and *Spp1* encoding Osteopontin. These proteins are associated with extracellular matrix (ECM) remodeling in skeletal muscle, that were suggested to play a role in the maintenance of sarcomere stability (Figure 2D). Differently, genes codifying for the atrophy-associated proteins Atrogin-1 (i.e., *Fbxo32*) and MuRF-1 (i.e., *Trim63*) were not modified by Ranolazine treatment (Figure 2D).

Additionally, since aged muscles display a low reservoir of skeletal muscle stem cells which also show an impaired myogenic potential [34], and since it has been found that a similar metabolic modulator triggers an increase in the number of centrally nucleated myofibers, we also decided to address this issue in the present study and we observed a remarkable increase in the percentage of centrally nucleated myofibers also in Ranolazine-treated mice (Figure 2E). In greater detail, we classified the centrally nucleated myofibers into fibers containing only one central nucleus (Figure 2F) and myofibers with two central nuclei (Figure 2G) and we observed an increased number of both classes of myofibers upon Ranolazine treatment, thus suggesting an increase in newly formed myofibers in vivo. This also correlates with the increased expression of the master gene of myogenesis *M**yod1* in Ranolazine-treated mice (Appendix A). In contrast, embryonic myosin heavy chain (eMyHC), Myogenin and Desmin transcript levels did not reveal any significant change (data not shown), although the stimulation of myogenesis had been previously reported in vitro [35]. We also performed Pax7 immunostaining and we did not observe any change in the number of satellite cells upon Ranolazine treatment (Appendix A), thus revealing no depletion of the muscle stem cell pool. This was also confirmed by the unchanged levels of *Pax7* transcripts (Appendix A).

Collectively, these histological and molecular data suggest an active skeletal muscle remodeling in aged mice upon Ranolazine treatment.

### 2.3. Ranolazine Up-Regulates Mitochondrial Genes and Boosts Antioxidant Enzyme Production Reducing Oxidative Stress in Skeletal Muscle of Aged Mice

Based on the knowledge that energy management is crucial for skeletal muscle and that oxidative metabolism seems to enhance myogenic potential [36,37,38,39], we analyzed the effect of Ranolazine on the expression of mitochondrial and oxidative metabolism markers. We found that the expression levels of the master regulator of mitochondrial biogenesis peroxisome proliferator-activated receptor-γ (PPARγ)-coactivator-1α (*Pgc-1*α), of cytochrome c (*Cytc*), of a complex I (*C-I*) subunit and also of a cytochrome-*c*-oxidase/complex IV (*C-IV*) subunit increase in the GSN of aged mice treated with Ranolazine compared to untreated controls (Figure 3A). We also analyzed the amount of mitochondrial DNA as an indicator of mitochondrial mass and we did not observe any difference in Ranolazine-treated muscle compared to untreated ones (data not shown), thus arguing against an increase in mitochondrial biogenesis triggered by Ranolazine. However, the detection of the respiratory chain gene over-expression observed in Figure 3A prompted us to analyze myofiber metabolism which we assayed by performing the in situ nicotinamide adenine dinucleotide dehydrogenase-tetrazolium (NADH-TR) activity assay which highlights the oxidative fibers. Consistently with the overexpression of respiratory chain genes, this assay revealed an increase in the percentage of an NADH-positive area, corresponding to a higher oxidative metabolism in Ranolazine-treated muscle compared to the untreated ones (Figure 3B). However, we did not observe any fiber type shift upon Ranolazine treatment as observed by the qPCR analysis of various MyHC transcripts (Appendix A). This is consistent with metabolic changes being more rapid than a complete shift to sarcomere components’ settlement.

We also analyzed the correlation between the CNFs and NADH-TR-positive fibers, and we observed that most of the CNFs we detected in Ranolazine-treated mice are NADH-TR-negative fibers. However, since the NADH-TR-highly positive fibers are lower in number, we measured the percentage of CNFs in negative-, low- and high-positive NADH-TR-fibers (Figure 3C). Collectively, the appearance of CNFs induced by Ranolazine is not specific of oxidative NADH-TR-positive myofibers (which are typically newborn myofibers); indeed we observed an increased percentage of CNFs in all fiber types, and mostly in NADH-TR-negative myofibers.

Finally, taking into consideration the fact that oxidative stress is another typical feature of aging [40], we analyzed the extent of skeletal muscle oxidative damage following Ranolazine treatment. We observed that lipid peroxidation, measured by 4-HNE labeling, decreases within muscles of Ranolazine-treated mice (Figure 4A). Moreover, we analyzed the nitrosative stress by the 3-NT labeling of skeletal muscle cryosections and we observed that Ranolazine strikingly reduces the 3-NT content compared to untreated old mice (Figure 4B). Based on these results, we evaluated, by qPCR, the expression levels of the nuclear factor erythroid 2-related factor 2 (*Nrf2*) gene, a transcription factor regulating the expression of many antioxidant cytoprotective genes including peroxiredoxin-2 (*Prdx2*) and heme oxygenase-1 (*Hmox1*), whose levels were also analyzed. From this analysis, we observed an increased expression in all three genes upon Ranolazine treatment (Figure 5A), thus confirming a role for this compound in protecting skeletal muscle from oxidative stress. In line with these results, we observed that Cu/Zn-SOD (*Sod1*), Mn-SOD (*Sod2*) and catalase (*Cat*)—all known antioxidant enzymes, some of which are reduced in aging sarcopenia [40]—were also up-regulated in Ranolazine-treated mice compared to control mice (Figure 5A). Consistently with these data, we performed dihydroethidium staining (DHE) on TA muscle sections, which allow the visualization and quantification of ROS levels. Our experiments revealed a strong decrease in the DHE-positive area in Ranolazine-treated mice, suggesting a reduced production of ROS and/or an increase in ROS scavenging upon Ranolazine treatment (Figure 5B). Overall, these data suggest a strong antioxidant effect of Ranolazine in old sarcopenic mice.

## 3. Discussion

In this study, we demonstrated that Ranolazine oral administration leads to a significant increase in muscle strength in aged mice, thus indicating that Ranolazine counteracts the functional impairment associated with age. This is in line with our previous results obtained by treating aged mice with another intraperitoneally administered metabolic modulator, Trimetazidine [28]. A deeper analysis of exercise capacity by treadmill would have corroborated the present data on muscle functionality. However, it would have required a separate group of animals since this kind of extreme exercise modifies the metabolic activity and the physiology of muscle which would have impaired the evaluation of the mere effect of Ranolazine. In contrast to Trimetazidine, here we also demonstrated that Ranolazine partially counteracts muscle mass loss in 24-month-old mice, as shown by the TA weight increase and as suggested by the trend of GSN weight to increase upon treatment. The trend of body weight decrease triggered by Ranolazine, and not paralleling the TA weight increase due to this drug, might depend on the strong fat mass decrease we observed—without quantifying it—during muscle collection in Ranolazine-treated mice. Consistently with this hypothesis, it has been reported that increased fat mass (sarcopenic obesity) might hide muscle depletion [41]. Further studies are required to confirm this hypothesis.

Our data show that the impact of Ranolazine on muscle functionality is stronger compared to its impact on muscle mass. This is consistent with the knowledge that muscle strength does not strictly correlate with muscle mass and CSA; indeed, it has been reported that muscle strength loss in the elderly mainly derives from impaired muscle quality and is approximately four-fold greater than muscle depletion [42]. Interestingly, we found that mRNA levels of *Postn* and *Spp1* are up-regulated by Ranolazine treatment. Periostin and Osteopontin are proteins important for sarcomere stability being involved in the costamere connection to the ECM. We can therefore speculate that the increased muscle functionality of old mice treated with Ranolazine and not associated with a parallel robust CSA increase might be related to an improvement in the sarcomere connection to the ECM. Consistently, we found that CNF are not specifically oxidative, which suggests that they are not newborn myofibers, but the increase in CNF is a general remodeling involving all fiber types. Additionally, the number of Pax7-positive cells, as well as the overall Pax7 and eMyHC gene expression, do not change following Ranolazine administration.

Our data also show that Ranolazine strongly protects skeletal muscle from oxidative damage by markedly decreasing ROS and RNS production, as revealed by the reduction in the DHE-positive area as well as by the decreased levels of 4-HNE and 3-NT which become comparable to the levels typical of young mice. This suggests an efficacious role of Ranolazine in reversing aging-related molecular mechanisms triggering oxidative stress and leading to functional decline. The beneficial impact of Ranolazine against oxidative stress has been previously reported in animal models that recapitulate cardiovascular diseases [43,44,45,46,47]. It has also been suggested that Ranolazine promotes myotube formation in vitro by counteracting oxidative stress [35]. ROS and RNS induce mitochondrial dysfunction and, consequently, muscle atrophy [7]. Being very close to the source of oxidants, and missing a robust repair system, mtDNA and mtDNA-dependent respiratory gene expression are very vulnerable to oxidative damage [7,48,49,50]. Vice versa, mitochondrial quality control mechanisms—among which are mitochondriogenesis and decreasing oxidative stress—maintain the integrity of mitochondria and consequently, improve skeletal muscle function [19,51]. Since our data show that Ranolazine leads to an up-regulation of mitochondrial markers and enhances the oxidative metabolism, we speculate that, by contributing to the clearance of ROS in myofibers, Ranolazine contributes to improving the mitochondrial metabolism which translates into an improvement in muscle functionality. This is also consistent with our recent data showing that chronic Ranolazine administration improves energy metabolism by increasing muscle ATP content and slows down muscle strength decline in a mouse model of amyotrophic lateral sclerosis (ALS) [52]. We must acknowledge some limitations in the execution of the present study; for technical reasons, however, we did not have the possibility to perform a polarographic measurement of mitochondrial respiration (oxygen consumption rate) on fresh muscles or on isolated mitochondria, which would have significantly improved the conclusions of this study. However, this point deserves further investigation.

Based on our data, we might speculate that the reduction in the oxidative stress might counteract mitochondrial decay, thus promoting energy generation which might be at the basis of the improved muscle performance induced by Ranolazine [7,8,53]. This hypothesis is also supported by data reported in the literature and suggesting that neuromuscular junction degeneration occurring in sarcopenia and impairing muscle strength might be caused by mitochondrial dysfunctions and low ATP synthesis [11,12,54,55,56]. Finally, we cannot rule out the possibility that the Na^+^ channel inhibitory role of Ranolazine and the subsequent facilitation of membrane repolarization might decrease the cell energy demand and compensate for mitochondrial failure typical of aging. Collectively, the improved muscle functionality of aged mice triggered by Ranolazine and its ability to counteract oxidative stress and to enhance the mitochondrial metabolism suggest a possible repositioning of this drug—which has already been approved for the clinical use—for the treatment of sarcopenia.

## 4. Materials and Methods

### 4.1. Ethical Approval

All animal procedures were conducted in accordance with the National Ethical Guidelines (Italian Ministry of Health; D.L.vo 26, 4 March 2014), approved by the local ethics committee (protocol number 375/2019/PR), and conformed to ARRIVE guidelines 2.0 to improve the reporting of research involving animals. The investigators declare that the principle of the “3R” (replace, reduce and refine) was also carefully fulfilled. Researchers were blinded to the experimental conditions and analyses.

### 4.2. Experimental Design

C57BL/6 male mice (24 months of age) (Charles River Laboratories, Wilmington, MA, USA) were utilized for the reported experiments. All mice considered for the experiments were 28–35 gr in weight; they were maintained on a 12 h/12 h light/dark cycle in temperature and humidity-controlled room with ad libitum access to food and water and were allowed to free cage activity. Based on prior analysis on old mice strength [28], a sample size, estimated by G*Power software (version 3.1.9.2.; Düsseldorf, Germany), of a minimum of 5 animals/group was calculated to be required to obtain significant results. Using a higher number of animals than that indicated by the sample size calculation would go against the principle of the “3R”. With respect to the previous work where the drug was given parenterally [52], in order to match the route of administration in patients, and given the pharmacokinetics of Ranolazine [57], the oral dose was increased and was set at 150 mg/kg. Mice were treated with 150 mg/kg/day Ranolazine (*n* = 5; Tocris, Bristol, UK) for 14 days by oral gavage, at 10–11 a.m.; age-matched control mice (*n* = 5) were administered with a vehicle (water). For the experiments, we included healthy mice with a body weight ranging from 28 to 35 g. Young male mice (11-week-old C57BL/6 mice; *n* = 4) were used as an additional control group. All the animals were sacrificed by cervical dislocation around 4–6 p.m. Afterwards, *tibialis anterior* (TA) and *gastrocnemius* (GSN) muscles were gently excised, rapidly collected and weighed. For histological studies, TA muscles were embedded in optimal cutting temperature (OCT) compound (Tissue Tek^®^, Sakura Finetek, Alphen aan den Rijn, The Netherlands) and then frozen in liquid nitrogen-cooled isopentane (Sigma-Aldrich, Merck KGaA, Burlington, MA, USA) for 20 s and then stored at −80 °C. GSN muscles were quickly snap frozen in liquid nitrogen for RNA extraction.

### 4.3. Functional Tests and Strength Measurement

Inverted screen test (or mesh test) was used to evaluate the muscle strength of the four limbs in rodents by measuring their latency to fall. We used the same protocol described in a previous paper before the sacrifice of mice [28]. Briefly, 24-month-old C57BL/6 male mice, treated or not treated with Ranolazine for 14 days, were individually placed at the center of a mesh screen (10 × 14 cm; wire thickness, 2 mm), then the grid was inverted upside-down with the mouse’s head declining first, and latency to fall off was recorded. The mesh screen was held steadily 50 cm above a padded surface to protect the mouse from injury. Mice were subjected to a 180 s hanging test, during which the longest time between two falls was recorded as the latency-to-fall value [32].

### 4.4. Immunofluorescence and Histological Analysis of Muscle

Embedded TA muscles were sectioned using a Leica cryostat (Leica Microsystems, CM1850UV, Wetzlar, Germany) set at −20 °C with a muscle section thickness of 8 μm. Afterwards, muscle sections were thawed and fixed in ice-cold acetone (Sigma-Aldrich, Merck KGaA, Burlington, MA, USA) for 1′, air-dried and washed: PBS was used for all washing steps. After blocking with 4% IgG-free BSA (Jackson ImmunoResearch, West Grove, PA, USA) in PBS for 45′, muscle slides were incubated overnight with anti-Laminin α1 antibody (1:500; Sigma-Aldrich Cat# L9393, RRID:AB_477163) at 4 °C. Samples were then washed with 1% IgG-free BSA and then incubated for 1 h with a goat anti-rabbit IgG (H+L), Alexa Fluor 488 antibody (1:500; Thermo Fisher Scientific, Waltham, MA, USA; Cat# A-11034, RRID:AB_2576217) at room temperature. Nuclei were counterstained with 4′,6 diamidino-2-phenylindole (DAPI, Thermo Fisher Scientific, Waltham, MA, USA; Cat# D1306, RRID:AB_2629482), washed twice and then mounted in 80% glycerol solution and a cover slide [58,59]. Representative images were acquired using an Olympus confocal microscope (Olympus FV1200, Olympus, Tokyo, Japan) with 40× magnification and visualized with FV10-ASW software (version 4.2; Olympus, Tokyo, Japan). Images for quantification were captured with a confocal laser scanning TCS SP5 microscope (Leica Microsystems, Wetzlar, Germany). Specifically, cross-sectional area (CSA) and centrally nucleated fibers (CNFs) were quantified from anti-Laminin α1/DAPI-stained muscle sections. Briefly, the analysis was performed using images of whole stained muscle sections, acquiring adjacent and non-overlapping fields (using 10× magnification). Thus, we quantified all muscle fibers of muscle sections corresponding to samples by using a semi-automated approach (modified from [60,61]. For fiber size distribution, muscle fibers were clustered in increasing ranges of 250 µm^2^ depending on the area of myofibers. Centrally nucleated fibers were quantified counting the number of DAPI-stained nuclei inside the muscle fiber which did not touch or overlap with the Laminin α1-positive signal. For Pax7 immunofluorescence, the antibody from Developmental studies Hybridoma bank (Iowa City, IA, USA; RRID:AB_528428) was used. Representative images were acquired using an Olympus confocal microscope (Olympus FV1200, Olympus, Tokyo, Japan) with 40× magnification and visualized with FV10-ASW software (version 4.2; Olympus, Tokyo, Japan).

NADH-TR staining was performed by incubating TA muscle cryosections for 40 min at 37 °C (in humidified incubation chamber), in NADH-TR working solution made of 0.4 mg/mL NADH, disodium, salt (Roche, Basel, Switzerland; Cat# 10107735001), 0.8 mg/mL Nitroblue tetrazolium chloride (NBT; Sigma-Aldrich, Merk KGaA, Burlington, MA, USA; Cat# N6876) and 0.1 M TRIS-HCl (Sigma-Aldrich, Merck KGaA, Burlington, MA, USA), pH 7.5. Slides were then washed twice with PBS, counterstained with DAPI, washed again and finally mounted in 80% glycerol solution. Representative images were acquired using an Olympus confocal microscope (Olympus FV1200, Olympus, Tokyo, Japan) with 40× magnification and visualized with the FV10-ASW software (version 4.2; Olympus, Tokyo, Japan). Images for the quantification of CNF fibers in NADH-TR-stained muscle sections were acquired with 20× magnification of 4 random and non-overlapping fields/per sample. Images for the quantification of the percentage of an NADH-positive area were acquired using an Olympus BX-41 microscope (Olympus, Tokyo, Japan) with 10× magnification and visualized using “cellSens Entry” software (version 3.1.1, Olympus, Tokyo, Japan).

Oxidative damage at the membrane level was determined by overnight incubation, at 4 °C, with anti-4-hydroxynonenal (4-HNE; 1:100; Thermo Fisher Scientific, Waltham, MA, USA; Cat# MA5-27570, RRID:AB_2735095). Peroxynitrite formation at amino acid sites was assessed by incubating anti-3-nitrotyrosine (3-NT; 1:100; Merck Millipore, Burlington, MA, USA; Cat# 06-284, RRID:AB_310089) antibody overnight at 4 °C. Tetramethylrhodamine B isothiocyanate-(TRITC)-conjugated phalloidin (1:200; Sigma-Aldrich, Merk KGaA, Burlington, MA, USA; Cat# P1951, RRID:AB_2315148) incubation, 1 h at 37 °C, was employed for staining actin filaments. Fluorescein isothiocyanate-(FITC; 1:100; Jackson ImmunoResearch Labs, West Grove, PA, USA; Cat# 115-095-003, RRID:AB_2338589) conjugated secondary antibody was used. Nuclei were counterstained with DAPI. Muscle sections were analyzed with an LSM700 (Zeiss, Oberkochen, Germany) confocal microscope [58].

The amount of ROS production was performed with dihydroethidium (DHE; Thermo Fisher Scientific, Waltham, MA, USA; Cat#: D11347) staining. Briefly, TA sections were thawed and incubated with 5 μM DHE in PBS, for 5 min at 37 °C in humidified incubation [62]. Nuclei were counterstained with DAPI. Images for histological analysis were acquired using motorized Leica LMD7000 microscope mounting a DFC345FX cam (Leica, Wetzlar, Germany) and using LAS X software (Leica, Wetzlar, Germany; RRID:SCR_013673). All histological analysis and quantifications were performed using ImageJ/Fiji (NIH, Bethesda, MD, USA; RRID:SCR_003070) software version 1.52t.

### 4.5. Quantitative PCR (qPCR) Analysis of Whole GSN Muscle

Total RNA from whole muscles was obtained by homogenizing the GSN muscle with a tissue homogenizer in TriReagent (Sigma-Aldrich, Merck KGaA, Burlington, MA, USA). RNA was extracted following TriReagent manufacturer’s protocol and then quantified with NanoDrop (Thermo Fisher Scientific, Waltham, MA, USA). For mRNA analysis, 500 ng of RNA was retro-transcribed with random primers. qPCR analysis was performed with 2X SYBR Green Master Mix (Applied Biosystems, Waltham, MA, USA). The primer sequences are as follows: ***Cat***: Fw-AGG TGT TGA ACG AGG AGG AG, Rv-AGG GTG GAC GTC AGT GAA AT; ***C-I***: Fw-TTG GGA ACA ACA GGA AGA GG, Rv-TTC CCA CTG CAT CCA TTA CA; ***C-IV***: Fw-CAG AAG GGA CTG GAC CCA TA, Rv-ATA ACA CAG GGG CTC AGT GG; ***Cytc***: Fw-GCC CGG AAC GAA TTA AAA AT, Rv-CCA GGT GAT GCC TTT GTT CT; ***Fbxo32***: Fw-AGC GCT TCT TGG ATG AGA AA, Rv-GGC TGC TGA ACA GAT TCT CC; ***Hmox1***: Fw-CCC ACC AAG TTC AAA CAG CTC, Rv-AGG AAG GCG GTC TTA GCC TC; ***Myh1***: Fw-GTT CCT CCT TCC AGA CCG TG, Rv-GGG GAT GAT GCA CCG TAC AA; ***Myh2***: Fw-TGC GGA ACT TGG ATA GAT TTG, Rv-TTG GTG GAT AAA CTC CAG GC; ***Myh4***: Fw-GTC CTT CCT CAA ACC CTT AAA GT, Rv-CAT CTC AGC GTC GGA ACT CA; ***Myh7***: Fw-TAC TCT GAC CAA GGC CAA GG, Rv-GCT GCT TGT CAT TCT CCA GG; ***Myod1***: Fw-GTC GTA GCC ATT CTG CCG, Rv-AGC ACT ACA GTG GCG ACT CA; ***Nrf2***: Fw-AGC AGG ACA TGG AGC AAG TT, Rv-TTC TTT TTC CAG CGA GGA GA; ***Pax7***: Fw-GCC GAG TGC TCA GAA TCA A, Rv-AGC CCT CAT CCA GAC GGT T; ***Pgc-1a***: Fw-TGA GGA CCG CTA GCA AGT TT, Rv-TGT AGC GAC CAA TCG GAA AT; ***Postn***: Fw-AAC CAA GGA CCT GAA ACA CG, Rv-AAC CCC ATT GGG ATA ATG GT; ***Prdx2***: Fw-ACT CTC AGT TCA CCC ACC TG, Rv-TAG TCA CGT CAG CAA GCA GA; ***Sod1***: Fw-AAC CAT CCA CTT CGA GCA GA, Rv-TAC TGA TGG ACG TGG AAC CC; ***Sod2***: Fw-ACG TGA ACA ATC TCA ACG CC, Rv-TGA AGA GCG ACC TGA GTT GT; ***Spp1***: Fw-GCT TGG CTT ATG GAC TGA GG, Rv-AGG TCC TCA TCT GTG GCA TC; ***Trim63***: Fw-ACC ACA GAG GGT AAA GAA GAA CA, Rv-GCA GAG AGA AGA CAC ACT TCC C; ***18S***: Fw-CGG AGA GGG AGC CTG AGA AAC, Rv-GTC GGG AGT GGG TAA TTT GCG. Quantitative PCR reactions were run on 7900HT ABI prism PCR machine (Applied Byosistems, Waltham, MA, USA). The reactions were run as technical duplicates and the output Ct were averaged for the analysis by standard ΔΔCt method. Data were reported as 2^−ΔΔCt^ × 10^5^, in order to uniformly express data by reducing decimal places. The expression level of selected targets was measured as relative to the housekeeping gene 18S.

### 4.6. Statistics

Data are presented as mean and standard deviation (SD). Statistical differences between groups were verified by Student’s *t*-test (2-tailed). Statistical differences for fiber size distribution were tested by ANOVA for repeated measures calculated using R Software (https://www.r-project.org/, accessed on 6 May 2022). *p*-values < 0.05 were considered statistically significant. Graphs were generated using GraphPad Prism (GraphPad Software, San Diego, CA, USA).

## 5. Conclusions

In conclusion, these data suggest an active skeletal muscle remodeling upon Ranolazine treatment, thus explaining the increased muscle force and confirming the beneficial effect of FDA-approved metabolic modulators on skeletal muscle, which paves the way for their use in sarcopenic elderly people. Moreover, based on our previous publication showing that Ranolazine slows down muscle strength decline in a mouse model of ALS [52], we propose that the metabolic modulators might represent valid drugs for counteracting skeletal muscle impairment, not only in the elderly population, but also in some pathological conditions characterized by skeletal muscle impairment, including ALS [63].

## Figures and Tables

**Figure 1 metabolites-12-00663-f001:**
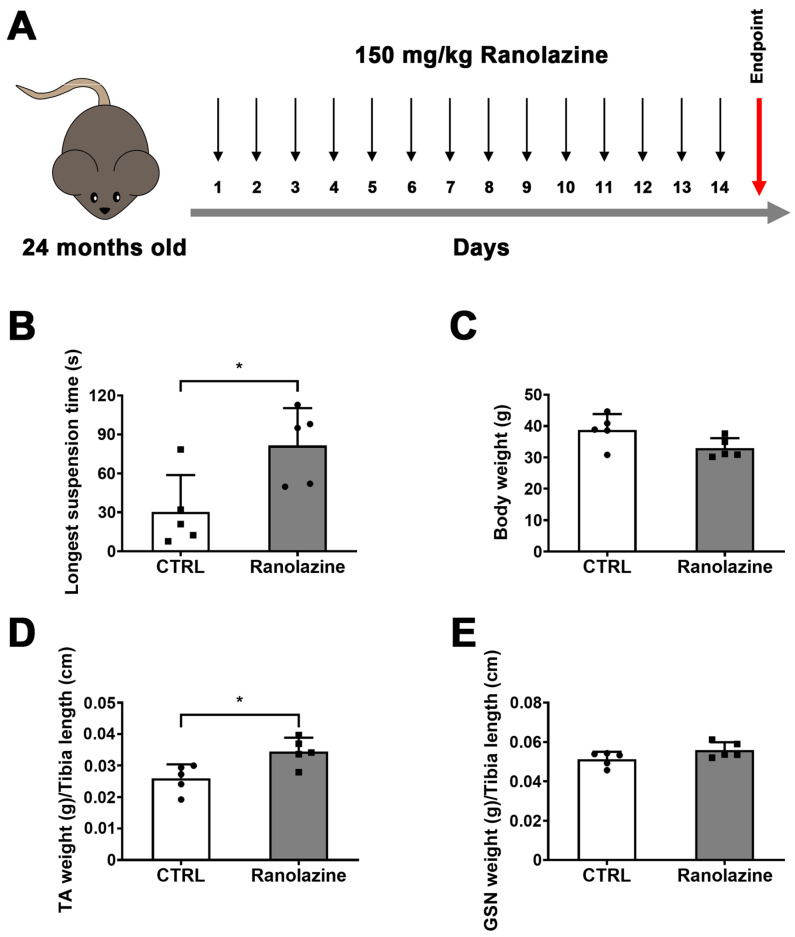
Fourteen-day-long treatment with Ranolazine increases the strength of old sarcopenic mice. (**A**) Experimental scheme for a 14-day-long Ranolazine treatment (by oral gavage) of 24-month-old C57BL/6 mice. Untreated mice were administered water. (**B**) Longest suspension time measured by inverted screen test (or mesh test) of untreated control (CTRL) mice and Ranolazine-treated mice. (**C**) Total body weight of mice included in the experiment and recorded before the sacrifice. (**D**,**E**) Weight of *tibialis anterior* (TA; (**D**)) and of *gastrocnemius* muscles (GSN; (**E**)) of untreated control (CTRL) mice and Ranolazine-treated mice normalized per tibia length. Values are means ± SD. *n* = 5 for both experimental conditions. Unpaired *t*-test was used for comparison. * = *p* < 0.05.

**Figure 2 metabolites-12-00663-f002:**
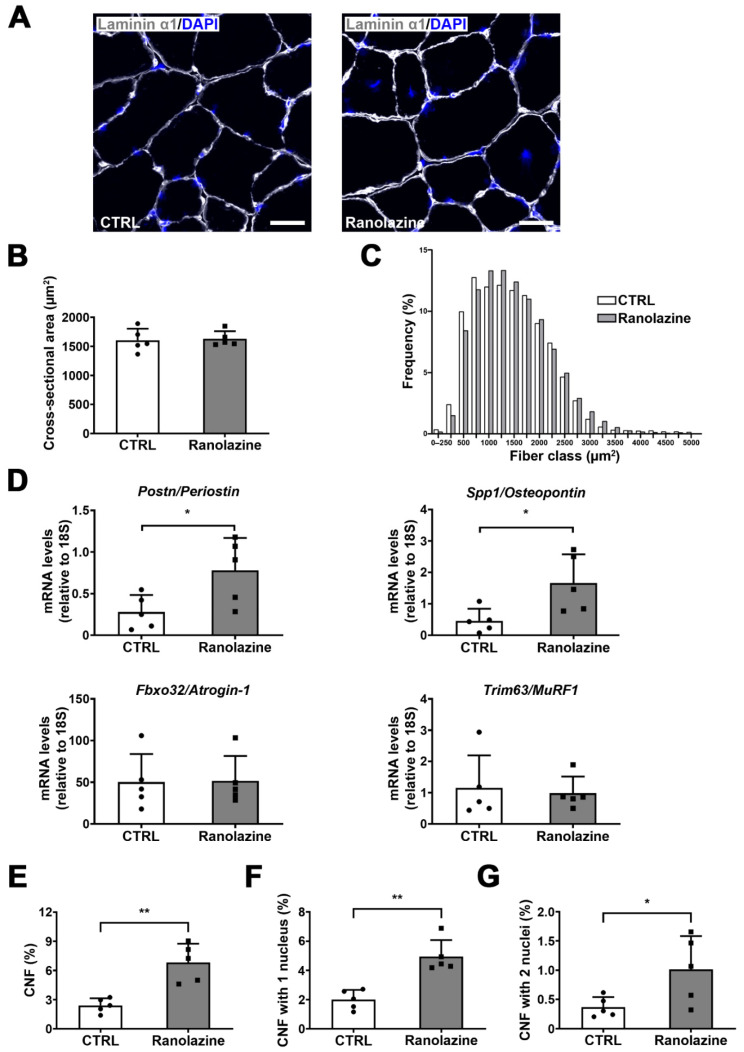
Ranolazine increases the number of centrally nucleated fibers and promotes muscle remodeling, without a marked effect on myofiber CSA. (**A**) Representative images of 8 µm-thick TA cryosections of untreated control (CTRL) mice and Ranolazine-treated mice stained with anti-Laminin α1 antibody (grey). Nuclei were counterstained with DAPI (blue). Scale bar = 25 µm. (**B**) Measurement of myofiber CSA in Laminin α1-stained TA cryosections shown in A. (**C**) A frequency histogram showing the distribution of myofiber CSA in Laminin α1-stained TA cryosections of untreated (CTRL) mice and Ranolazine-treated mice (F (1, 8) = 0.426; *p* = 0.532). (**D**) qPCR expression analysis of *Postn* (Periostin), *Spp1* (Osteopontin), *Fbxo32* (Atrogin-1) and *Trim63* (MuRF1) on whole TA muscles derived from old mice treated or not treated with Ranolazine. Data are reported as relative to housekeeping gene 18S. (**E**–**G**) Percentage of centrally nucleated fibers (CNFs) in Laminin α1- and DAPI-stained TA cryosections expressed as total CNFs (**E**), CNFs with only 1 nucleus (**F**) and CNFs with 2 nuclei (**G**) in untreated (CTRL) and Ranolazine-treated mice. Values are means ± SD. *n* = 5 for both experimental conditions. Unpaired *t*-test was used for comparison. ** = *p* < 0.01, * = *p* < 0.05.

**Figure 3 metabolites-12-00663-f003:**
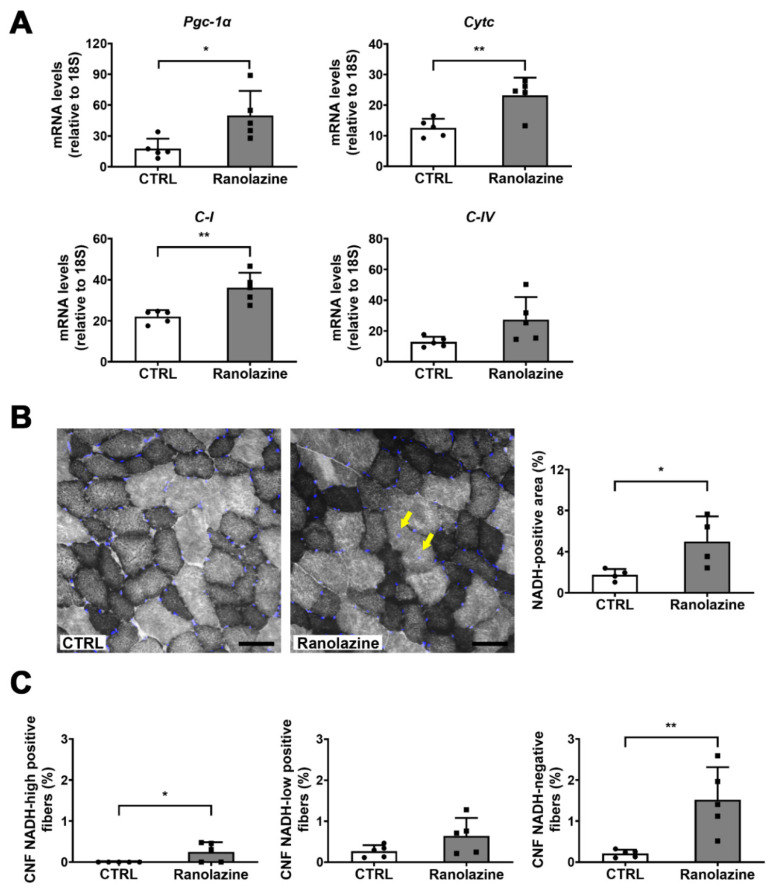
Up-regulation of mitochondrial genes and increase in NADH-dehydrogenase upon Ranolazine treatment. (**A**) qPCR expression analysis of key mitochondrial genes on whole TA muscle lysates of untreated (CTRL) and Ranolazine-treated mice. All data are reported as relative to the housekeeping gene 18S. Values are means ± SD. *N* = 5 for both experimental conditions. (**B**) Representative images of NADH-TR staining in cross-sections of TA muscle of 24-month-old mice treated with Ranolazine or not (CTRL). Nuclei were counterstained with DAPI (blue). Yellow arrows point at CNF. The histogram shows the percentage of the NADH-TR positive area of whole TA section. Scale bar = 50 μm. Values are means ± SD. *n* = 4 for both experimental conditions. (**C**) Percentage of CNF NADH-TR-high positive, CNF NADH-TR-low positive and CNF NADH-TR-negative fibers. Values are means ± SD. *n* = 5 for both experimental conditions. Unpaired *t*-test was used for comparison. ** = *p* < 0.01, * = *p* < 0.05.

**Figure 4 metabolites-12-00663-f004:**
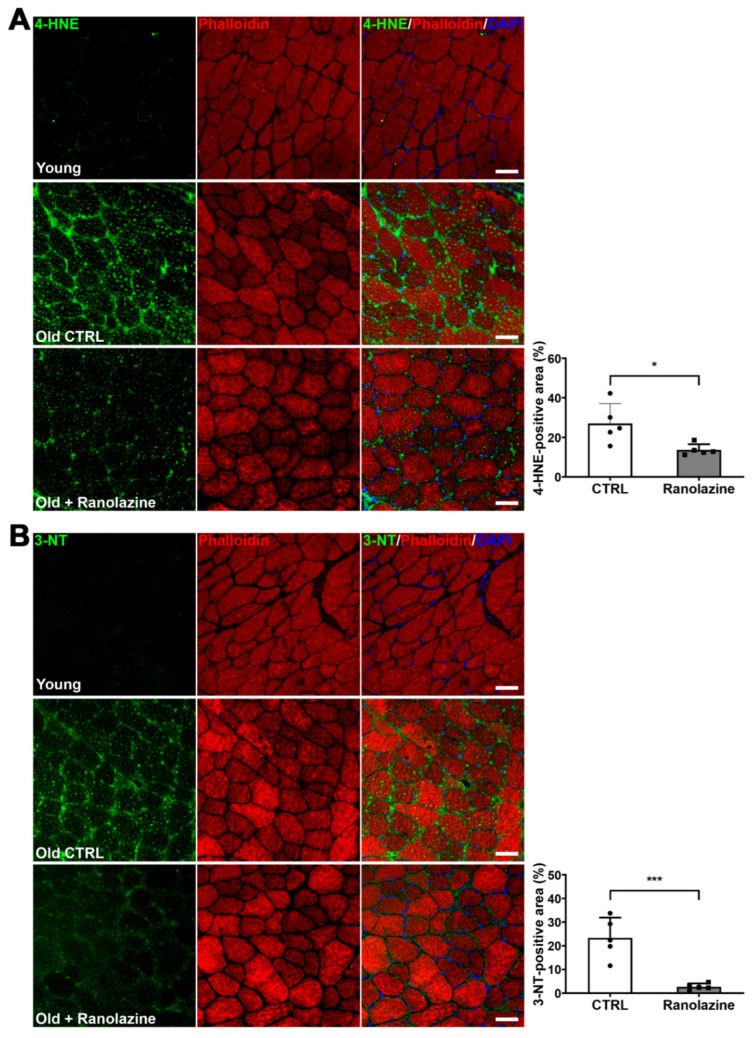
Ranolazine strongly reduces oxidative and nitrosative stress in old sarcopenic mice. (**A**) Representative images of 8 µm-thick TA cryosections of young mice and old mice treated or not treated (Old CTRL) with Ranolazine, stained with an anti-4-Hydroxynonenal (4-HNE) antibody (green) and TRITC-conjugated phalloidin (red) to stain the cytoplasm. Nuclei were counterstained with DAPI (blue). Scale bar = 50 µm. The histogram displays the quantification of the percentage of the 4-HNE-positive area. (**B**) Representative images of 8 µm-thick TA cryosections of young and old mice treated or not treated (Old CTRL) with Ranolazine, stained with an anti-3-nitrotyrosine (3-NT) antibody (green) and TRITC-conjugated phalloidin (red) to stain cytoplasm. Nuclei were counterstained with DAPI (blue). Scale bar = 50 µm. The histogram displays the quantification of the percentage of the 3-NT-positive area. Values are means ± SD. *n* = 5 for both experimental conditions. Unpaired *t*-test was used for comparison. *** = *p* < 0.001, * = *p* < 0.05.

**Figure 5 metabolites-12-00663-f005:**
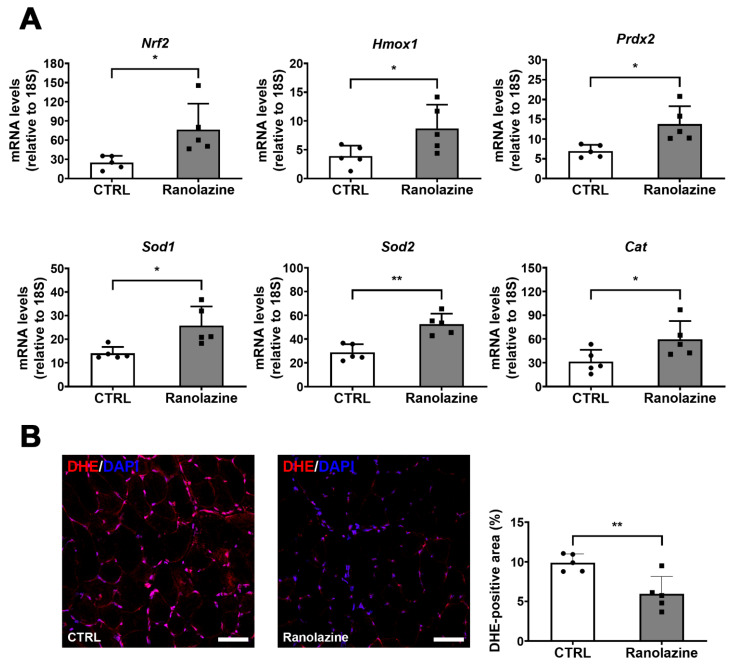
Up-regulation of antioxidant genes and decreased ROS in skeletal muscle upon Ranolazine treatment. (**A**) qPCR expression analysis of key genes involved in oxidative stress response on whole TA muscle lysates derived from old mice treated with Ranolazine or not (CTRL). Data are reported as relative to the housekeeping gene 18S. (**B**) Representative images of 8 µm-thick TA cryosections of old mice treated with Ranolazine or not (CTRL) stained with Dihydroethidium (DHE, red). Nuclei were counterstained with DAPI (blue). Scale bar = 50 µm. The histogram displayed the quantification of the percentage of DHE-positive area. Values are means ± SD. *n* = 5 for both experimental conditions. Unpaired *t*-test was used for comparison. ** = *p* < 0.01, * = *p* < 0.05.

## Data Availability

The data presented in this study are available in the article and Appendix A.

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
