# Peer review of "Ranolazine Counteracts Strength Impairment and Oxidative Stress in Aged Sarcopenic Mice"

_metabolites, 2022, doi:10.3390/metabo12070663_

Round 1
Reviewer 1 Report
In this article, Torcinaro & Capetta et al. investigate the role of a Ranolazine treatment on skeletal muscle of aged sarcopenic mice. This manuscript highlights the protective role of Ranolazine against oxidative stress in aged skeletal muscles. Overall, the manuscript is clearly written and the work is well done. However, several points need to be addressed:
Major comments:
- The dose of Ranolazine used in this study is 150mg/kg. However, in the previous study on ALS, Scaricamazza et al. stated that “50 mg/kg of RAN was selected as the most effective dose.” In addition, the mode of administration is also different. The reasons why the authors decided to use a higher dose of Ranolazine given by gavage in this study should be justified.
- A major point concerns the origin of the increase in muscle mass. The authors show that there are no differences in CSA despite the increase in muscle mass in the Ranolazine-treated mice. However, they observe an increase in the number of CNF, which in itself cannot explain the effect on muscle mass. How did they quantify CSA? The methods used for all histological quantifications are not described in the materials and methods for all quantifications. How many fibers did they quantify? The measurement of all muscle fibers by automatic analysis (published in many papers) may increase the statistical significance of the results. They did observe a change in the frequency of CSA fibers classes but statistics are lacking. The authors should strengthen these aspects of the paper and also maybe analyze the expression of genes/proteins associated with muscle hypertrophy/atrophy.
- Ranolazine treatment in aged mice induces an increase in NADH-positive fibers. Is this related to changes in fibre typing? Is there a correlation between the appearance of NADH-positive fibres and CNF?
- The increase in the number of CNF in Ranolazine-treated mice and the increase in Myod1 expression suggest that Ranolazine activates quiescent Pax7-positive muscle stem cells. Many studies have shown that the activation of these cells is sensitive to oxidative stress/ROS modulation. The authors need to verify whether this treatment activates muscle stem cells and depletes the muscle stem cell pool. Measurement of total Pax7 expression by RT-qPCR as well as quantification of Pax7+ and Myod+ cells after treatment should be performed to analyze the side effects of Ranolazine treatment on the muscle stem cell pool. This is important since the depletion of the muscle stem cell pool could counteract the positive effects of this treatment on skeletal muscle.
Minor comments:
- References 6 and 17 do not fit the sentences that refer to them.
- Results showing an increase in body fat in mice treated with Ranolazine should be included in the paper.
- Scale bars are not visible on many images.
- Because of the importance of circadian rhythms in skeletal muscle function, the authors could specify what time of day Ranolazine was given and when the mice were sacrificed.
- Some of the information in the introduction, especially about mitochondrial metabolism, is redundant. These parts can be synthesized.
- In the discussion, it would be interesting to compare the differences between the Trimetazine and Ranolazine treatments in skeletal muscle as differences exist.
Author Response
Major comments:
Q1
The dose of Ranolazine used in this study is 150mg/kg. However, in the previous study on ALS, Scaricamazza et al. stated that “50 mg/kg of RAN was selected as the most effective dose.” In addition, the mode of administration is also different. The reasons why the authors decided to use a higher dose of Ranolazine given by gavage in this study should be justified.
A1
Yes, we should have explained our choice. For this study, we have decided to administer ranolazine by oral gavage to match the route of administration in patients. Moreover, as experienced in our study on trimetazidine the daily i.p. injection is stressful for aged mice. Starting from a previous study when ranolazine was administered by parenteral route (50mg/kg i.p.), the daily oral dose had to be adjusted for pharmacokinetics differences between the two routes. Generally, to produce a comparable pharmacological effect, higher doses are required when a drug is administered orally (compared to ip). This is true also for ranolazine, whose oral bioavailability ranges from 35% to 50% (Jerling M. Clinical Pharmacokinetics of Ranolazine. Clin Pharmacokinet 2006). A comment has been included in the Methods.
Q2
A major point concerns the origin of the increase in muscle mass. The authors show that there are no differences in CSA despite the increase in muscle mass in the Ranolazine-treated mice. However, they observe an increase in the number of CNF, which in itself cannot explain the effect on muscle mass. How did they quantify CSA? The methods used for all histological quantifications are not described in the materials and methods for all quantifications. How many fibers did they quantify? The measurement of all muscle fibers by automatic analysis (published in many papers) may increase the statistical significance of the results. They did observe a change in the frequency of CSA fibers classes but statistics are lacking).
A2
We have now described in detail the method we have used to quantify CSA. We have measured 1100-2000 myofibers (on the entire muscle section) per animal (n=5 CTRL mice and n=5 Ranolazine-treated mice).
As specified, we have observed a trend to an increase in the number of fibers with higher caliber upon Ranolazine treatment. However, this difference is not significant (we now added the statistics). We can only record a trend which would require a much higher number of animals to be confirmed or rejected.
Methodsi: Cross-sectional area (CSA) and centrally-nucleated fibers (CNF) were quantified from anti- Laminin/DAPI-stained muscle sections. Briefly, the analysis was performed using images of whole stained muscle sections, acquiring adjacent and non-overlapping fields. Thus, we quantified all muscle fibers of muscle sections corresponding to samples, by using a semi-automated approach (modified from J Muscle Res Cell Motil 2006;27(1):1-8. doi: 10.1007/s10974-005-9014-9; Masschelein et al. Skeletal Muscle (2020) https://doi.org/10.1186/s13395-020-00237-2).
Q3a
The authors should strengthen these aspects of the paper and also maybe analyze the expression of genes/proteins associated with muscle hypertrophy/atrophy.
A3a
We thank this Reviewer for his suggestion which allowed us to improve our manuscript. We have performed a qPCR analysis of some genes found to be upregulated in muscle growth and thought to play a role in the maintenance of sarcomere. Among them, genes codifying the protein involved in atrophy Atrogin-1 (i.e. Fbxo32) and MURF-1 (i.e. Trim63) were not modified by RAN treatment, whereas RAN induced an increase of Postn encoding Periostin and Spp1 encoding Osteopontin. These proteins are necessary for muscle growth and, in particular, for ECM remodeling that play a role in the maintenance of sarcomere stability (PMID: 25554798). These data are now reported in Fig 2D.
Q3b
Ranolazine treatment in aged mice induces an increase in NADH-positive fibers. Is this related to changes in fiber typing?
Is there a correlation between the appearance of NADH-positive fibres and CNF?
A3b
Again, we need to thank this Reviewer for his suggestion. We performed qPCR of MyHC (Myosin heavy chain) isoforms to evalutate fiber types and we did not observe any difference between RAN-treated and untreated OLD mice (Fig S1D). We commented that metabolic changes are quicker than changes in gene expression and typology of fibers rearrangement.
We also analyzed the correlation between the NADH-positive fibers and CNF, and we observed that most of the CNF we observed in RAN-treated mice belong to the NADH-negative fibers, with no evident correlation between the appearance of NADH-positive fibers and CNF. However, since the NADH-positive fibers are low in number, we also measured the percentage of CNF in high, medium and negative NADH fibers (Fig 3B, C ). On the whole, upon Ranolazine treatment we observed and increase of CNF in all fiber types. Therefore, the appearance of CNF is not specific of NADH-positive fibers but it is a remodeling involving all fiber types.
Q4
The increase in the number of CNF in Ranolazine-treated mice and the increase in Myod1 expression suggest that Ranolazine activates quiescent Pax7-positive muscle stem cells. Many studies have shown that the activation of these cells is sensitive to oxidative stress/ROS modulation. The authors need to verify whether this treatment activates muscle stem cells and depletes the muscle stem cell pool. Measurement of total Pax7 expression by RT-qPCR as well as quantification of Pax7+ and Myod+ cells after treatment should be performed to analyze the side effects of Ranolazine treatment on the muscle stem cell pool. This is important since the depletion of the muscle stem cell pool could counteract the positive effects of this treatment on skeletal muscle.
A4
To answer this question, we performed an immunostaining for Pax7 (Fig S1B). We did not observe any difference in the amount of Pax7 positive cells upon Ran treatment and also by qPCR (Fig S1B,C). Although we have tested 4 different antibodies raised against MyoD, we have not succeeded in performing MyoD staining. However, Pax7 data show that there is not influence of Ranolazine on the muscle stem cell pool. The up-regulation of Myod (now Fig S1A) could suggest a remodeling of muscle but without affecting the satellite cells.
Minor comments:
Q1-References 6 and 17 do not fit the sentences that refer to them.
A1-We apologize for such a confusion. We removed old ref 6 and 17 which should have been the following:
We changed ref 6 with:
De Santa, L. Vitiello, A. Torcinaro, and E. Ferraro, "The Role of Metabolic Remodeling in Macrophage Polarization and Its Effect on Skeletal Muscle Regeneration," (in eng), Antioxid Redox Signal, Oct 2018, doi: 10.1089/ars.2017.7420.
We changed Ref 17 with the following:
Ferraro E, Giammarioli AM, Chiandotto S, Spoletini I, Rosano G. Exercise-Induced Skeletal Muscle Remodeling and Metabolic Adaptation: Redox Signaling and Role of Autophagy. Antioxid Redox Signal. 2014 Jul 1;21(1):154-76. doi: 10.1089/ars.2013.5773.
Q2- Results showing an increase in body fat in mice treated with Ranolazine should be included in the paper.
A2- We apologize for not being clear on this topic. Unfortunately, we have not performed a measurement of fat mass. We have only qualitatively reported a strong reduction of white adipose tissue which we did not measure. For this reason, we changed the sentence “The trend of body weight decrease triggered by Ranolazine -and not paralleling the TA weight increase triggered by this drug- might depend on the strong fat mass decrease we observed in Ranolazine-treated mice (data not shown)” with the following one: “The trend of body weight decrease triggered by Ranolazine -and not paralleling the TA weight increase triggered by this drug- might depend on the strong fat mass decrease we have observed -without quantifying it- during muscle collection in Ranolazine-treated mice.
We moved these comments on the discussion paragraph.
Q3-Scale bars are not visible on many images.
A3 – We modified them
Q4-Because of the importance of circadian rhythms in skeletal muscle function, the authors could specify what time of day Ranolazine was given and when the mice were sacrificed.
A4- As stated by the Reviewer, skeletal muscle function is subjected to circadian rhythm. The two-week treatments (ranolazine or vehicle) were done in the morning, 10-11am. Mice were sacrificed in the afternoon, around 4-6 pm.
Q5-Some of the information in the introduction, especially about mitochondrial metabolism, is redundant. These parts can be synthesized.
A5-Following this Reviewer suggestion, we synthetized this part.
Q6-In the discussion, it would be interesting to compare the differences between the Trimetazine and Ranolazine treatments in skeletal muscle as differences exist.
A6-Thanks for this suggestion. We completed the comparison between the effect of ranolazine and trimetazidine in aged mice, both in the discussion and in some paragraphs.of the results.
Reviewer 2 Report
Authors Torcinaro et al., examined ranolazine’s function in muscle tissue in sarcopenic mice. This is a continued work by the group following the study of Trimetazidine muscle wasting caused by aging. Ranolazine and Trimetazidine are both metabolic modulator of beta-oxidation. Through sarcopenic mice (24mo-old) experiments (14-day treatment), Authors reported that ranolazine improved muscle strength, up-regulate antioxidant and mitochondrial genes. They concluded the ranolazine protects muscle from oxidative stress. The data demonstrated by the manuscript are well presented and support the conclusion. Below are some points for authors to address:
1. The strength measurement presented in 4.3 only include an inverted screen test. Is there any other measurement data authors can provide? For example force measurement.
2. For figure 2A the laminin staining seems to be blur in the ranolazine group. Was that an artifact? Or Ranolazine affects the ECM?
3. For result session 2.3, no western blotting of mitochondria related proteins were shown. Authors should consider adding some western results.
Author Response
Authors Torcinaro et al., examined ranolazine’s function in muscle tissue in sarcopenic mice. This is a continued work by the group following the study of Trimetazidine muscle wasting caused by aging. Ranolazine and Trimetazidine are both metabolic modulator of beta-oxidation. Through sarcopenic mice (24mo-old) experiments (14-day treatment), Authors reported that ranolazine improved muscle strength, up-regulate antioxidant and mitochondrial genes. They concluded the ranolazine protects muscle from oxidative stress. The data demonstrated by the manuscript are well presented and support the conclusion. Below are some points for authors to address:
Q1
The strength measurement presented in 4.3 only include an inverted screen test. Is there any other measurement data authors can provide? For example force measurement.
A1
As this Reviewer commented, “this is a continued work following the study of Trimetazidine on muscle wasting caused by aging”. We followed the same structure as our previous work which shows that data obtained by the inverted screen test are analogous to those obtained by grip test. We also found that the inverted screen test is more stringent than the grip test (REF 28). Therefore, we considered the data obtained by the inverted screen test highly reliable to highlight a difference in strength.
Q2
For figure 2A the laminin staining seems to be blur in the ranolazine group. Was that an artifact? Or Ranolazine affects the ECM?
A2- We thank this Reviewer for addressing this point. We have replaced representative images for both CTRL and Ranolazine experimental conditions (FIG 2A). We decided to use 40X X2 zoomed images instead of 40X images which were digitally 2X-zoomed.
Q3
For result session 2.3, no western blotting of mitochondria related proteins were shown. Authors should consider adding some western results.
A3
We thank this Reviewer for his suggestion that, we know, would have improved the quality of our manuscript. We also had all the tools to perform western blots. However, we experienced some technical troubles which led to the degradation of our RIPA lysates. We hope that NADH-TR stainings, the immunofluorescences and the other experiments we have performed might be considered by this reviewer as a partial answer to his question.
Round 2
Reviewer 1 Report
The authors have carefully addressed all concerns I had.